# Price elasticity of demand for ready-to-drink sugar-sweetened beverages in Brazil

**Auberth Henrik Venson[1], Larissa Barbosa Cardoso[2], Flaviane Souza Santiago[3]\*, Kênia Barreiro de Souza[4], Renata Moraes Bielemann[5]**

**1** Department of Economics, State University of Londrina, Londrina, Paraná, Brazil, **2** Faculty of Administration, Accounting and Economic Sciences, Federal University of Goiás, Goiânia, Goiás, Brazil, **3** Faculty of Economics, Federal University of Juiz de Fora, Juiz de Fora, Minas Gerais, Brazil, **4** Department of Economics, Federal University of Parana, Curitiba, Paraná, Brazil, **5** Faculty of Nutrition, Federal University of Pelotas, Pelotas, Rio Grande do Sul, Brazil

\* santiago.flaviane@ufjf.br

**Data Availability Statement:** All data files are available from the Household Budget Survey (POF 2017/2018). (Available from: https://www.ibge.gov.br/estatisticas/sociais/rendimento-despesa-e-

## Abstract

The taxation of sugar-sweetened beverages is a policy that has been adopted in many countries worldwide, including Latin American, to reduce sugar consumption. However, little is known about how taxation on these products may affect their demand. The present study aims to estimate the price elasticity of demand for sugar-sweetened beverages in Brazil. This study advances the literature by proposing a breakdown between ready-to-drink sugar-sweetened beverages and sugar-sweetened beverages that require some preparation before being consumed. With this disaggregation, it is possible to obtain more accurate elasticities for the group of products that will be effectively taxed. We estimated a Quadratic Almost Ideal Demand System (QUAIDS) model using the Household Budget Survey 2017–2018 microdata. The results show that ready-to-drink beverages is more consumed but less sensitive to changes in price than prepared beverages. The price elasticity of demand for ready-to-drink and prepared sugar-sweetened beverages was -1.19 and -3.38. Additionally, we observe heterogeneity in these price elasticities across household incomes, with a more elastic demand among lower-income households for ready to drink beverages. The findings suggest that taxing ready-to-drink sweetened beverages could potentially reduce sugar consumption directly through a decrease in the consumption of sugary drinks and this effect could be reinforced by reducing the consumption of other sugar-rich products. Therefore, the taxation police should effective contribute to minimize health risks associated to the sugar consumption.

## 1. Introduction

In recent decades, the prevalence of non-communicable diseases (NCDs) has been increasing worldwide, posing a challenge in public health policies. The growth in the incidence of these NCDs in low- and middle-income countries represents an obstacle to poverty reduction due to the loss of productivity related to premature deaths [1, 2]. Among all deaths caused by non-communicable diseases (NCDs), 77% occur in low- and middle-income

consumo/9050-pesquisa-de-orcamentos-familiares.html?=&t=o-que-e).

**Funding:** Regarding financial support, we are grateful for the financial support of the National Council for Scientific Development and Tecnológico (CNPq) for the development of research. The funders had no role in study design, data collection and analysis, decision to publish, or preparation of the manuscript.

**Competing interests:** The authors have declared that no competing interests exist.

countries. In Brazil, specifically, the incidence of NCDs has been rising and accounted for 75.9% of the total deaths in 2019 [3].

The consumption of sugar-sweetened beverages (SSB) has been emphasized as one of the main risk factors for NDCs [4]. Extensive literature highlights this relationship and highlights SSBs as one of the main sources of added sugar in the diet [5–9]. SSB is associated with greater risk of premature death and a significant burden of disease in developed and developing countries [10, 11]. It has been estimated that globally, the consumption of sugar-sweetened beverages (SSBs) could be linked to 184,000 deaths per year [12].

The prevalence of SSB consumption is increasing globally, especially in Latin America. Estimates show that the average daily SSB consumption per adult in this region is three times higher than the global average. In 2015, four of ten countries with the highest consumption of sugar-sweetened beverages were in Latin America [13]. Brazil figures among these countries, with 14% of adults consuming SSB regularly in 2021 [14]. Even though the consumption has reduced in the last decade (from 23% to 15.4%), the mean per capita intake of soda in Brazil continues to be high (67.7 ml/day) [15]. Furthermore, with a tropical climate and a sizable population, Brazil presents an enticing opportunity to expand the sales of SSB.

Due to the association between SSB consumption and NCDs, reducing its consumption of these beverages has been discussed as an alternative to improve health outcomes. The World Health Organization (WHO) has recommended implementing SSB taxation policies to discourage the consumption of SSB [16]. Taxation should contribute to increasing the SSB prices and, consequently, reduce its consumption. Studies endorse taxation policies showing that the SSB tax reduces the consumption of SSB, increases the consumption of healthy beverages, and is cost-effective for dealing with the growth in the intake of high-sugar foods [17]. However, the effectiveness of these policies is affected by various factors such as pass-through rate, and own- and cross-price elasticities demand [18, 19].

Price elasticities of demand for sweetened beverages reveal how the consumption of these products would behave in the face of taxation. Studies that sought to estimate the price elasticity of demand for sugar-sweetened beverages in Latin American countries identified an elastic demand. Considering the various countries and varying aggregations of SSBs among these studies, the price elasticity of demand averaged approximately -1.26, within a range of -0.71 to -1.73 [20–28]. Systematic review studies have emphasized that there is still little literature available that focuses mainly on estimating the elasticities of demand for sweetened beverages in Latin American countries [17, 29].

Similar results are observed in Brazil, where evidence show that SSB tax can reduce the consumption [26, 28, 30]. The most studies focus in ready-to-drink sugar-sweetened beverages. Equally, the agreement signed by the Ministry of Health and companies in the food and beverage sector has considered those ready for consumption beverages, including sodas, soft drinks, and nectars [31]. However, beverages prepared, such as powdered drinks and beverages concentrated in liquid, may also contain sugar preparation [32]. The process of preparation, whether through dilution or the addition of other ingredients, can difficult the accuracy of the quantity of added sugar effectively consumed. The consumption of these beverages tends to be lower in Brazil (one-fourth of soda consumption in 2017–2018) [33]. However, the substitution of ready-to-drink with beverages prepared may occur in a SSB tax scenario and limit the effects of these policies.

In this context, the present study aims to estimate the price elasticity of demand for sugar-sweetened beverages in Brazil, considering ready-to-drink beverages and sweetened beverages for preparation disaggregation. Furthermore, we examined the variations in elasticities across households with different income levels, specifically comparing the poorest and wealthiest households.

The is organized as follow. The section 2 presents a brief discussion of the literature that focuses on the estimation of the price elasticity of demand for sweetened beverages in Latin American countries. Section 3 provides the methodology used by this study, with the database, variables, and the statistical model. The results obtained are presented and discussed in section 4, followed by the final considerations.

## 2. Literature review

This section presents a brief review of studies whose focus is the estimation of the price elasticity of demand for sugar-sweetened beverages in Latin American countries. The studies generally use demand system estimation methods to obtain the elasticities, highlighting the variations of the Almost Ideal Demand System (AIDS) and Quadratic Almost Ideal Demand System (QUAIDS) models as the main estimation techniques.

In Mexico, the price elasticity of demand for sugar-sweetened beverages was estimated using an LA/AIDS model to analyze the effect of taxation on these products. The SSB was divided into soft drinks and other sweetened beverages. Both categories of sweetened beverages showed elastic demand, with these elasticities being higher for lower-income strata and rural families or those living in marginalized municipalities [20].

For Ecuador, the estimation of price elasticity of demand for sugar-sweetened beverages sought to identify possible substitute goods using the NL/AIDS model. The results showed that the demand for sugar-sweetened beverages in Ecuador is elastic and the cross-elasticity showed that unsweetened beverages are substitutes for sugar-sweetened beverages. Based on income elasticity, sugar-sweetened beverages were considered superior goods for lower-income classes and normal goods for higher-income classes [21].

The price elasticities of demand for sugar-sweetened beverages in Chile, separating sugar-sweetened beverages into the categories of soft drinks and other sweetened beverages, sought to analyze the cross elasticities of sugar-sweetened beverages with other high-energy foods (such as sweets and snacks), for which the LA/AIDS and QUAIDS models were estimated. In both models, the demand for soft drinks and other sugar-sweetened beverages proved to be elastic, with the elasticity of other sweetened beverages being greater than the elasticity of soft drinks, and according to the cross elasticities, all other product categories showed a behavior of being substitute goods for soft drinks [22].

In Guatemala, the estimated own-price elasticity, cross-elasticity, and expenditure elasticity for sugar-sweetened beverages and other beverages using an NL/AIDS model identified that the demand for soft drinks was elastic, with greater elasticity for rural families, and the demand proved to be inelastic for urban families. All beverages considered (soft drinks, packaged juices, milk, and bottled water) proved to be normal goods. The cross elasticities, on the other hand, presented ambiguous results regarding the statistical significance of these estimates. This model included a variable to measure household food security and identified that higher consumption of sugar-sweetened beverages is related to food insecurity [23].

Estimated elasticities of demand for sugar-sweetened beverages in Argentina divided sugar-sweetened beverages into two categories: soft drinks and juices and flavored waters, for which an NL/AIDS model was estimated. The results indicated that the demands for both categories of sugar-sweetened beverages were elastic and had very close elasticity values. The two categories of sugar-sweetened beverages were identified as normal goods and proved to be substitute goods for each other [24].

The elasticity of demand for sugar-sweetened beverages estimated in Ecuador using a QUAIDS model divided sugar-sweetened beverages into two categories: soft drinks and other sugar-sweetened beverages. The results showed that the demands for the two categories of

sugar-sweetened beverages were elastic, and the two categories of sugar-sweetened beverages proved to be complementary goods. Coffee and tea as substitute goods for sugar-sweetened beverages [25].

For Brazil, the first study that focused on the estimation of elasticities of demand for sugar-sweetened beverages sought to estimate the own-price elasticity of demand for sugar-sweetened beverages based on data from the Household Budget Survey (HBS) for the years 2002–2003. The demand for sugar-sweetened beverages was about identified as inelastic [26]; however, these results were obtained through the estimation of an OLS model, which is a limited method for calculating elasticities compared to demand systems estimation methods. In the context of demand system estimation methods, the most recent studies [27, 28], which estimated a QUAIDS model for calculating the elasticities of demand for sugar-sweetened beverages in Brazil, stand out.

The estimated demand for sugar-sweetened beverages in Brazil based on HBS data from 2008–2009 analyzed the elasticities of 4 categories of sugar-sweetened beverages: cola soft drinks, other soft drinks, energy drinks, and industrialized juices. The results showed an inelastic demand for cola soft drinks and energy drinks and an elastic demand for other soft drinks and industrialized juices. It was also identified that cakes and sweets and other processed foods were considered complementary goods to cola and other soft drinks, while milk, tea, coffee, and dairy drinks stood out as substitute goods for cola soft drinks and other soft drinks [27].

The elasticities of demand for sugar-sweetened beverages in Brazil estimated based on HBS data from 2017–2018, the demand for soft drinks, sugar-sweetened beverages based on milk, chocolate, or soy and other sugar-sweetened beverages proved to be elastic. Milk, natural juice, coffee, and tea proved to be substitute goods for sugar-sweetened beverages, and it is also worth noting that these beverage categories were considered substitutes for each other [28].

Studies in general indicate a trend of elastic demand for sugar-sweetened beverages in Latin American countries. The studies differ in the variations of the AIDS and QUAIDS models used, and the disaggregation used to categorize the group of sugar-sweetened beverages. The present study's main difference is in the breakdown of sweetened beverages between ready-to-drink beverages and those for preparation, a breakdown that was not utilized in any of the studies discussed. This disaggregation between these two categories is appropriate because the taxation on sugar-sweetened beverages aimed at reducing consumption focuses, in general, on ready-to-drink beverages. This approach also considers the possibility of substitution between ready-to-drink beverages and those for preparation. Thus, the estimated elasticities with this disaggregation are more accurate in terms of measuring the effect of taxation on the consumption of sugar-sweetened beverages.

## 3. Materials and methods

### 3.1 Data and variables

To estimate the elasticities of demand for SSB in Brazil, we used the microdata from national representative *Pesquisa de Orçamentos Familiares* (POF–Household Budget Survey) of 2017–2018 [34] collected by the Brazilian Institute of Geography and Statistics (IBGE). The HBS-IBGE used a complex cluster sampling procedure, drawn from two-stage stratification process. The first stage selected the census tracts, while the second stage selected the households within those tracts [33]. Data were collected from between July 11, 2017 to July, 2018 using a group of questionnaires.

For this study, we used the information from the POF-2 questionnaire (Collective Acquisition Notebook), which includes information on purchases of food, beverages, cleaning

products, fuel for domestic use, and other products whose purchases serve all residents [34, 35]. We focused on sugar-sweetened beverages and classified these beverages in ready-to-drink sweetened beverages and prepared sugar-sweetened beverages Ready-to-drink SSB includes soda, nectar, and juice. For their part, prepared sugar-sweetened beverages (powdered refreshments and concentrated beverages). Furthermore, we take into consideration other food categories to compose the estimated demand system and consider the replacement of sweetened beverages with foods high in sugar and/or calories [36]. These groups include diet soda; whole juice; dairy beverages; energy drinks; milk; coffee and tea; water; ice cream; candy; snacks and pizza; bakery; and other foods. The expenditure and the quantity acquired for each product were recorded daily for seven consecutive days. All expenditure values were adjusted for inflation as of January 15, 2018 [33]. The quality-adjusted unit value was considered as the average unity price calculated as the expenditure divided by total quantity consumed in the category, weighted by expenditure share [37]. For non-consuming households, the missing unit values were replaced with the regional average price of a group commodity in the corresponding state [38].

The household characteristics and socioeconomic data of the residents were obtained from POF 1 questionnaire. The group of characteristics included in the analysis is composed by dichotomous variable for the area (rural and urban), dichotomous variable for the region (Northeast, North, Midwest, Southeast and South), the natural logarithm of household income, gender and race of the head of household, education attained by the head of household in years, household structure (number of rooms, sewage, piped water), five dichotomous variable for number of residents by age group, dichotomous variable for credit card and dichotomous variable for oven, stoven, refrigerator and freezer.

The demand system was estimated from observations with complete data for all product groups considered, and the final model was then estimated from a sample with 47,261 observations.

### 3.2 Statistical analysis - Quadratic almost ideal demand system

Own- and cross-price elasticities (measures the percentage change in demand for a product in response to a percentage change in its own price and in other prices, respectively) and expenditure elasticities for ready-to-drink and prepared SSBs were estimated by running Quadratic Almost Ideal Demand System (QUAIDS) [39]. QAIDS allows for flexible price responses, allows for nonmonotonic income effects and heterogenous cross-price elasticities. The demand system estimated by the QUAIDS model is defined as:

$$w_i = \alpha_i + \sum_{j=1}^{n} \gamma_{ij} ln p_j + \beta_i \ln\left(\frac{m_h}{a(\boldsymbol{p})}\right) + \frac{\lambda_i}{b(\boldsymbol{p})}\left[\ln\left(\frac{m_h}{a(\boldsymbol{p})}\right)\right]^2 + \varepsilon_i \qquad (1)$$

where $w_i$ is the expenditure share of product $i$; $p_j$ is the price of product $j$; $m_h$ is the total food expenditure for household $h$; and $b(p)$ and $a(p)$ are, respectively, the Matsuda price index and Tornqvist price index [40]; $\alpha_i$, $\gamma_{ij}$, $\beta_i$ and $\lambda_i$ are the model parameters and $\varepsilon i$ is the error term.

The price variables of the products used in the model were obtained through the average unit value of each product in the Brazilian state, thus all households faced positive prices for all products in the system, even if there are no expenses with any of the products in the system. household, and there is price variability between households, as households located in different states will face different prices. See S1 Table for details of the descriptive statistics of prices and quantities for each food category.

Some households did not recorded expenses on SSB and this case may produce biased estimates [25, 27, 41]. To take this in consideration, we performed the two-step procedure developed by Shonkwiler and Yen [42] for estimating a system of equations with limited dependent variables. The first step consists of estimating probit models of the consumption decision of each product category, controlling for Tornqvist price index and sociodemographic variables (see the S2 Table).

In the second step, we calculated the cumulative distribution functions (cdf) and probability density functions (pdf). Then, we included it in the estimation of the demand system to correct bias caused by the zero-consumption problem. The QUAIDS model estimated in the second step of the procedure then becomes:

$$w_{ih} = \Phi_i(\mathbf{z}_h)\left\{\alpha_i + \sum_{j=1}^{n}\gamma_{ij}lnp_j + \beta_i\ln\left(\frac{m_h}{a(\boldsymbol{p})}\right) + \frac{\lambda_i}{b(\boldsymbol{p})}\left[\ln\left(\frac{m_h}{a(\boldsymbol{p})}\right)\right]^2\right\} + \rho\varphi_i(\mathbf{z}_h) + \varepsilon_i \quad (2)$$

where $\Phi_i(\mathbf{z}_h)$ e $\phi_i(\mathbf{z}_h)$ are, respectively, the cdf and pdf obtained through the estimated probit model for the product $i$ e $\mathbf{z}_h$ is the vector of sociodemographic variables used to estimated probit models.

We adopted the procedure proposed by Blundell e Robin [43] to accounting for endogeneity problem of total expenditure on food. In this case, conducted a regression analysis using a set of independent variables that represent household characteristics and calculated the residuals of the regression. The residuals were included as independent variable in the QUAIDS model. After performing this procedure, the final estimated QUAIDS model was defined as:

$$w_{ih} = \Phi_i(\mathbf{z}_h)\left\{\alpha_i + \sum_{j=1}^{n}\gamma_{ij}lnp_j + \beta_i\ln\left(\frac{m_h}{a(\boldsymbol{p})}\right) + \frac{\lambda_i}{b(\boldsymbol{p})}\left[\ln\left(\frac{m_h}{a(\boldsymbol{p})}\right)\right]^2 + \theta_i\hat{v}_h\right\} + \rho_i\phi_i(\mathbf{z}_h)$$
$$+ \xi_i \quad (3)$$

where $\alpha_i$, $\gamma_{ij}$, $\beta_i$, $\lambda_i$, $\theta_i$ e $\rho_i$ are the parameters of the QUAIDS model and $\xi_i$ is the error term.

After correcting the problems of zero consumption and endogeneity of total food expenditure, the price elasticities and expenditure elasticity is calculated by differentiating the Eq (3). The price elasticities, obtained by the delta method, are given by:

$$e_{ij} = \Phi_i(\mathbf{z}_h)\left\{\frac{\gamma_{ij} + \mu_i(\alpha_i + \sum_{k=1}^{K}\gamma_{kj}lnp_k) - \frac{\lambda_i\beta_i}{b(\boldsymbol{p})}\left[\ln\left(\frac{m_n}{a(\boldsymbol{p})}\right)\right]^2}{w_i}\right\} - \delta_{ij} \quad (4)$$

where $\mu_i = \beta_i + 2\lambda_i\ln\left(\frac{m_h}{a(\boldsymbol{p})}\right)$ and $\delta_{ij}$ is the Kronecker delta, which assumes a value of 1 for own-price elasticity or a value of 0 for cross-price elasticity. The food expenditure elasticity on food was also calculated, which is given by:

$$e_i = \Phi_i(\mathbf{z}_h)\left\{\frac{\beta_i + \frac{2\lambda_i}{b(\boldsymbol{p})}\ln\left(\frac{m_h}{a(\boldsymbol{p})}\right)}{w_i}\right\} + 1 \quad (5)$$

To present the income differences in own-price elasticity we estimated the elasticities for the lower income (first quintile) and higher income (fifth quintile) households. All parameters were estimated by the nonlinear seemingly unrelated regression (NLSUR) method using STATA 14.

**Table 1. Proportion of families that consumed, average expenditure, and share of income spent on average with the products–Household Budget Survey 2017–2018 - Brazil.**

| Product Categories | Families that consumed (%) | Average Expenditure (R$) | Share of Total Food Expenditure (%) |
|---|---|---|---|
| Ready-to-Drink SSB | 24.56 | 2.60 | 2.27 |
| Diet soda | 0.96 | 0.09 | 0.04 |
| Whole Juice | 5.13 | 0.85 | 0.51 |
| Prepared SSB | 15.78 | 1.29 | 0.91 |
| Dairy Beverages | 20.28 | 2.10 | 1.43 |
| Energy Drinks | 0.83 | 0.09 | 0.05 |
| Milk | 41.56 | 5.37 | 4.95 |
| Coffee and Tea | 28.09 | 3.46 | 2.54 |
| Water | 7.14 | 0.67 | 0.75 |
| Ice Cream | 2.88 | 0.46 | 0.29 |
| Candy | 17.02 | 1.95 | 1.16 |
| Snacks and Pizza | 9.75 | 1.65 | 1.13 |
| Bakery | 32.55 | 2.97 | 2.72 |
| Other Foods | 98.95 | 105.21 | 81.21 |

Source: Prepared by the authors

## 4. Results and discussion

Table 1 presents a brief description of the consumption of the product categories considered in the estimation of the QUAIDS model. The results reveals that 24.56% of the surveyed households consumed ready-to-drink sweetened beverages and spend on average R$2.60 on theses beverages committing 2.27% of their income. For prepared sweetened beverages, 15.78% of households consume this type of beverage and spend R$1.29 on average. It was noted that the consumption of ready-to-drink sweetened beverages has a prevalence of almost nine percentage points higher than the consumption of beverages for preparation and that the average expense and share of income spent was more than twice as high for ready-to-drink beverages than for beverages for preparation. It should also be noted that 6.26% of families consume both type of beverages.

Among the other categories of products included in the demand system, milk, bakery, and coffee and tea stand out in terms of consumption. Milk was the product with the highest occurrence of consumption, with 41.56% of families having consumed the product; it was also the product with the highest average expenditure and the largest share of income spent on average. This was followed by the categories of Bakery and Coffee and Tea, which were, respectively, the second and third places in terms of consumption, average expenditure, and share of income spent on average.

Table 2 presents the descriptive statistics of SSB consumption, ready-to-drink SSB, and prepared SSB, by household characteristics. The prevalence of ready-to-drink sugar-sweetened beverage purchases in Brazilian households was higher than that for prepared SBB (24.56% and 15.78%, respectively). The percentages for ready-to-drink SSBs were higher in the Southeast Region, urban areas, households with income of 10 or more minimum wage salaries, households with 4 or 5 residents, and households with children or adolescents. For prepared SSB, the prevalence of purchase shows similar results. Furthermore, the prevalence of prepared SSBs was monotonic with income and number of residents, while for ready-to-drink SSBs has the same behavior for income, the prevalence assumes an inverted U-shape for number of

**Table 2. Prevalence of SSB purchase, average expenditure on SSB, conditional average expenditure, and proportion of income spent on SSB by type of SSB and subgroups.**

| | n | Prevalence of SSB purchase (%) | Average expenditure (R$) | Conditional average expenditure (R$) | Conditional Share of total food expenditure spent on SSB (%) | Prevalence of SSB purchase (%) | Average expenditure (R$) | Conditional average expenditure (R$) | Conditional Share of total food expenditure spent on SSB (%) |
|---|---|---|---|---|---|---|---|---|---|
| | | | Ready-to-drink SSB | | | | SSB prepared | | |
| **Total sample** | 47261 | 24.56 | 2.60 | 10.60 | 9.24 | 15.78 | 1.29 | 8.23 | 5.74 |
| **Region** | | | | | | | | | |
| North | 3383 | 20.57 | 1.85 | 8.99 | 8.69 | 11.45 | 0.62 | 5.36 | 4.28 |
| Northeast | 13100 | 16.77 | 1.43 | 8.56 | 8.71 | 9.13 | 0.53 | 5.82 | 4.84 |
| Midwest | 3511 | 25.50 | 2.72 | 10.68 | 10.28 | 17.19 | 1.50 | 8.72 | 6.29 |
| Southeast | 19981 | 27.09 | 3.11 | 11.49 | 9.21 | 19.30 | 1.74 | 9.08 | 6.08 |
| South | 7282 | 33.01 | 3.60 | 10.91 | 9.58 | 19.31 | 1.64 | 8.51 | 5.73 |
| **Location** | | | | | | | | | |
| Urban | 40753 | 25.87 | 2.75 | 10.65 | 9.25 | 16.71 | 1.39 | 8.34 | 5.85 |
| Rural | 6508 | 16.38 | 1.63 | 10.09 | 9.18 | 9.97 | 0.71 | 7.10 | 4.55 |
| **Income** | | | | | | | | | |
| 0 to 2 MWS | 10879 | 15.62 | 1.21 | 7.74 | 11.86 | 11.20 | 0.54 | 4.73 | 6.05 |
| 2 to 4 MWS | 15308 | 22.06 | 2.04 | 9.28 | 9.88 | 13.39 | 0.89 | 6.43 | 5.57 |
| 4 to 10 MWS | 14892 | 30.37 | 3.45 | 11.38 | 9.03 | 17.86 | 1.64 | 9.03 | 5.79 |
| 10 MWS or more | 6186 | 32.45 | 4.39 | 13.54 | 6.44 | 21.56 | 2.78 | 12.53 | 5.62 |
| **Number of residents** | | | | | | | | | |
| 1 resident | 6153 | 15.41 | 1.40 | 9.07 | 11.80 | 10.70 | 0.74 | 6.93 | 7.61 |
| 2 to 3 residents | 24902 | 24.62 | 2.58 | 10.52 | 9.21 | 15.11 | 1.24 | 8.22 | 6.09 |
| 4 to 5 residents | 13195 | 28.53 | 3.14 | 11.03 | 9.07 | 18.55 | 1.63 | 8.83 | 5.19 |
| 6 or more resident | 3010 | 25.35 | 2.81 | 11.10 | 7.25 | 19.66 | 1.41 | 7.25 | 4.40 |
| **Residents under 18 years old** | | | | | | | | | |
| Yes | 23054 | 27.02 | 2.83 | 10.50 | 9.29 | 18.77 | 1.47 | 7.85 | 5.24 |
| No | 24207 | 22.21 | 2.38 | 10.72 | 9.20 | 12.95 | 1.13 | 8.74 | 6.42 |

Source: Own elaboration based on data from HBS 2017/18. Note: SSB–sugar-sweetened beverages; MWS–minimum wage salaries

residents, which indicates a reduction of the prevalence of purchase in the higher categories of this variable.

In terms of average expenditure (Table 2), this difference is not observed, and the pattern is similar between both types of SSB. On average, the expenditure on ready-to-drink SSBs and prepared SSBs was R$2.60 and R$1.29. Households in the South Region, urban areas, with high income, more residents, and with the presence of children and adolescents spent more on these beverages. There is heterogeneity within the subgroups, and in most of them, the highest expense corresponds to twice the value of the category with the lowest expense. When we considered only the households that consume SSBs, represented by conditional average expenditure, the observed inequality in expenditure decreases. There is a smaller difference in spending between the different subgroups of the sample. Furthermore, the conditional average

expenditure reveals that among those who consume these beverages, the average expenses on SSBs are 4 higher, and the monetary difference is greater among those with higher income.

Considering the expenses on SSBs in terms of household total food expenditure, we observed that Brazilian households spend most of their income on ready-to-drink SBBs. This corresponds to 9.24% of their household total expenditure, while spending on prepared SSBs represents 5.74%. The poorest household spent 11.86% of their total food expenditure on ready-to-drink SSBs, compared with 6.44% spent by the wealthiest households. In terms of prepared SSBs, the results reveal a smaller difference, with the poorest household spending 6.05% of their total food expenditure while the wealthiest households spent 5.62%. A greater share of income is also spent among households in the Midwest Region, although spending on SSBs in this region is not the highest. The results also show that in households with only one resident, the percentage of income spent on both types of SSB is higher than in those with a greater number of residents.

Table 3 presents the own- and cross-price elasticities of the selected food groups. Overall, we found that SSBs are price elastic. The results indicate a price elasticity of demand of -1.19 for ready-to-drink sugar-sweetened beverages. This result suggests that by increasing the ready-to-drink SSB price by 20%, we would observe a reduction of 23.8% in purchases of this type of beverage. This result is similar those from elasticities of demand for SSBs estimated for other Latin American countries [21–24], and the results found in previous studies for Brazil [27, 28].

For the prepared SSBs, the results show an elasticity of -3.38, which was higher than that observed for ready-to-drink SSBs. This suggests that the demand for prepared SSB is more sensitive to changes in price, and the consumption of this type of SSB could be more affected by an eventual SSB tax. Furthermore, the cross-price elasticity between these two types of SSB was negative, which suggests complementarity between these beverages. The existence of complementarity between different groups of sugar-sweetened beverages has been pointed out in another study [25]. Thus, a tax on ready-to-drink beverages, in addition to leading to a reduction in their consumption, also leads to a drop in the consumption of prepared SSBs.

Considering that the objective of SSB tax is to improve health by reducing the consumption of sugar, other food groups would also be affected by this policy. The cross-price elasticities show that an increase in the prices of ready-to-drink SSBs is associated with a decrease in the consumption of foods that are high in added sugar, such as sweets and ice cream; and greater consumption of other beverages such as energy drinks, whole juice, milk and dairy beverages. This result is consistent with those found in the literature, which pointed out sweets as complementary goods for sugar-sweetened beverages [20, 27, 36]. The substitution relationship between ready-to-drink SSBs and whole juice has already been highlighted by previous studies in Brazil [27, 28]. For prepared SSB, we found that a price increase for this type of beverage is associated with a smaller consumption of whole juice, dairy beverages, and sweets.

The elasticity results show that a tax on ready-to-drink SSBs may potentially lead to a reduction in the population's consumption of sugar directly, with a drop in the demand for this product, and indirectly, with a drop in the demand for complementary foods, such as sweets and sweetened beverages for preparation. An association is also observed with the high consumption of dairy beverages, energy drinks and milk. In addition to price elasticities, Table 4 presents the estimates of total food expenditure elasticities. The results show an elasticity of expenditure of -0.51 for ready-to-drink SSBs and -0.01 for prepared SSBs. This indicates that increases in household total food expenditure are associated with smaller consumption of ready-to-drink SSBs and prepared SSBs, however, the magnitude of the expenditure elasticity for prepared SSBs is economically unimportant. The results obtained in previous studies identified sugar-sweetened beverages as normal goods in other Latin American countries [21, 23, 24], which brings a difference in what was found in the present study. The difference observed

**Table 3. Own and cross-price elasticity of the demand for SSB and other.**

| | Price elasticity | | | | | | | | | | | | | |
|---|---|---|---|---|---|---|---|---|---|---|---|---|---|---|
| | **Ready to drink SSB** | **Diet Soda** | **Whole Juice** | **Prepared SSB** | **Dairy Beverages** | **Energy drink** | **Milk** | **Coffee and tea** | **Water** | **Ice Cream** | **Sweets** | **Snacks and Pizza** | **Bakery** | **Other foods** |
| Ready to drink SSB | **-1.19***** | 0.00 | 1.57*** | -0.38*** | 0.66*** | 0.25*** | 1.11*** | -0.35*** | -0.56*** | -1.09*** | -0.32*** | -0.33 | -0.96*** | -1.77*** |
| Diet Soda | -0.03 | **1.92** | 1.51*** | -0.14 | -0.12 | 0.46* | -0.26 | 0.03 | -0.43*** | -0.26 | -0.10 | 1.22 | -2.75*** | 0.21*** |
| Whole Juice | 0.16*** | 0.10*** | **-1.24***** | -0.04*** | 0.02*** | 0.02*** | -0.05 | -0.02 | -0.10*** | 0.39*** | 0.00 | 0.04*** | -0.11*** | -0.50** |
| Prepared SSB | -0.59*** | -0.12 | -0.46*** | **-3.38***** | -0.34*** | 0.19*** | 0.06 | 0.10 | 0.23*** | 0.07 | -0.27*** | 1.60*** | -0.68*** | 0.03*** |
| Dairy Beverages | 0.95*** | -0.08 | 0.41*** | -0.29*** | **-1.68***** | 0.10** | -0.25*** | -0.58*** | -0.28*** | -0.51*** | -0.81*** | 0.16 | -0.34** | -3.40*** |
| Energy drink | 0.32*** | 0.35* | 0.24*** | 0.15*** | 0.09** | **-1.06** | -0.10 | -0.14*** | 0.11** | 1.13*** | -0.02 | -0.66*** | -0.18*** | 0.10*** |
| Milk | 1.03*** | -0.07 | 0.02 | 0.03 | -0.07*** | -0.03 | **-1.84***** | 0.00 | -0.44*** | 0.60*** | 0.17 | 0.19*** | 0.23*** | 1.25*** |
| Coffee and tea | -0.23*** | 0.05 | 0.00 | 0.06 | -0.42*** | -0.10*** | -0.06 | **-2.20***** | 0.36*** | -0.35*** | 0.01*** | 0.75*** | 0.05 | -0.56 |
| Water | -0.28 | -0.10*** | -0.35*** | 0.04*** | -0.12*** | 0.04** | -0.41*** | 0.07*** | **-1.58***** | -0.05 | -0.04* | 0.17*** | 0.26*** | -1.24*** |
| Ice Cream | -0.31 | 0.03 | 0.76*** | -0.01 | -0.12*** | 0.19*** | 0.03*** | -0.15*** | -0.05 | **-1.81***** | 0.11 | -0.02 | 0.14** | -1.42*** |
| Sweets | -0.37*** | -0.06 | 0.07 | -0.21*** | -0.80*** | -0.01 | 0.06 | 0.23*** | 0.12* | -0.06 | **0.11** | 0.02 | -0.19* | 0.22*** |
| Snacks and Pizza | 0.21 | 0.49*** | 0.31*** | 0.65*** | 0.01 | -0.32*** | 0.13*** | 0.48*** | 0.31*** | 0.02 | 0.00 | **-3.49***** | -0.42*** | -0.17*** |
| Bakery | -0.84*** | -1.48*** | -0.71*** | 0.38*** | -0.24*** | -0.12*** | 0.13*** | -0.02 | 0.74*** | 0.92*** | -0.16* | -0.58*** | **-3.44***** | -0.90*** |
| Other foods | 1.27*** | 0.31*** | 1.20*** | 0.03 | -0.57*** | 0.25*** | 0.57*** | 0.07 | 1.34*** | 2.17*** | 0.14*** | 0.38*** | 0.82*** | **-12.83***** |

Source: Own elaboration based on data from HBS 2017/18.

**Note:** *** $p < 0.01$

** $p < 0.05$

* $p < 0$.

in the expenditure elasticity of ready-to-drink beverages and prepared beverages reinforce the importance of separating these two categories for analyzing the demand for sweetened beverages, as it shows a difference in the consumption pattern of these groups of products.

The income groups' own-price elasticities of demand for ready-to-drink SSB were also calculated, shown in Table 5. Analyzing the own-price elasticity by income groups we verified that the demand is elastic among families in the first income quintile, and for families in the fifth income quintile, the demand for ready-to-drink SSB becomes inelastic.

This result shows that lower-income families are more sensitive to changes in the price of ready-to-drink SSB than higher-income families, similarly to what was identified for Mexico [20]. Thus, a tax on sweetened beverages would have a greater potential to reduce consumption among poorer families than richer families in Brazil.

## 5. Conclusion

Many countries in Latin America have recently adopted taxes on sweetened beverages to reduce the consumption of sugar and, consequently, to prevent the associated negative health outcomes. Thus, the present study sought to estimate the elasticity of demand for sugar-

**Table 4. Total food expenditure elasticities estimated by the QUAIDS model.**

| Food Category | Estimates |
|---|---|
| Ready-to-drink SSB | -0.51*** |
| Diet Soda | -0.10*** |
| Juice | -0.25*** |
| Prepared SSB | -0.01** |
| Dairy Beverages | -0.14*** |
| Energetics | -0.08*** |
| Milk | 0.08*** |
| Coffee and Tea | -0.07*** |
| Water | -0.43*** |
| Ice cream | -0.44*** |
| Candies | -0.02*** |
| Snacks and Pizza | -0.13*** |
| Bakery | -0.28*** |
| Other foods | 2.28*** |

Source: Own elaboration based on data from HBS 2017/18.

Note: *** $p < 0.01$

** $p < 0.05$

* $p < 0.1$

sweetened beverages in Brazil, disaggregating SSBs into ready-to-drink sugar-sweetened beverages and sugar-sweetened beverages for preparation.

This categorization was adopted because the taxation on SSBs is generally focused on sodas, soft drinks, and nectars. The separation into beverages for preparation and ready-to-drink beverages brings greater accuracy to the elasticities calculated to assess the effects of a possible tax on sugar consumption in general since the concentration of sugar in ready-to-drink beverages is already defined, while the concentration of sugar in beverages for preparation may vary

**Table 5. Price elasticity in relation to changes in ready to drink SSB prices for first quintile and fifth quintile of income–Household Budget Survey 2017–2018 –Brazil.**

| Food Category | Quantile 1 | Quantile 5 |
|---|---|---|
| Ready to drink SSB | -1.12 | -0.56 |
| Diet Soda | -1.93 | 0.15 |
| Whole Juice | -0.03 | 0.63 |
| Prepared SSB | 0.28 | -0.71 |
| Dairy Beverages | 1.76 | 0.99 |
| Energy drink | 0.29 | 0.13 |
| Milk | 0.91 | 0.78 |
| Coffee and tea | -0.36 | -0.17 |
| Water | -0.30 | -0.26 |
| Ice Cream | -1.19 | -0.97 |
| Sweets | -0.94 | -0.22 |
| Snacks and Pizza | 1.05 | -0.31 |
| Bakery | -2.12 | -0.20 |
| Other foods | 1.17 | 0.57 |

Source: Own elaboration based on data from HBS 2017/18.

depending on their dilution. Additionally, possibilities of substitution between the groups are considered.

The results showed that the demand for sugar-sweetened beverages in Brazil is elastic, enabling the tax policy effectiveness. It was also observed that the price elasticity of demand for sugar-sweetened beverages for preparation is greater than the elasticity of those that are ready to drink. However, the main point of the disaggregation is to obtain the cross-price elasticity between ready-to-drink SSB and those for preparation. Our results show that ready-to-drink sugar-sweetened beverages and sugar-sweetened beverages for preparation are complementary goods, i.e., the rising prices of ready-to-drink sugar-sweetened beverages reduce the demand for sugar-sweetened beverages for preparation.

Thus, a tax on ready-to-drink sugar-sweetened beverages has the potential to reduce the consumption of sugar by the Brazilian population, in addition to reducing its demand. This relationship becomes clearer when considering that the group of sweets also proved to be a complementary good for ready-made sweetened drinks. Therefore, the implementation of a tax on ready-to-drink sweetened beverages, such as soft drinks or nectars, would have a direct effect on reducing the population's sugar consumption by reducing the demand for the product itself, as well as an indirect effect by reducing the demand for sweets and sweetened beverages for preparation. It should be noted that this effect of reduction in consumption may be even greater among poorer families than richer families, given that families with lower incomes showed a more elastic demand than families with higher income levels.

Considering that, a taxation on sweetened beverages can contribute to improving health outcomes by increasing the price of these beverages and reducing consumption. With a higher sensitivity among lower-income households than higher-income households, the taxation of SSBs has the potential to promote health equity, especially in terms of the prevalence of non-communicable chronic diseases.

One of the strengths of this research lies in its implications for food policy. For a more comprehensive understanding of the implementation of these measures in Brazil, future studies should calculate the net benefits of SSB taxation, assessing both the health benefits and consumer welfare losses. It's important to acknowledge some limitations of the study. Firstly, it exclusively focuses on food expenses, neglecting other types of household expenditures. The constraints in quantifying these other items prevent the calculation of implicit prices, consequently impeding the estimation of a more comprehensive demand system. Additionally, the estimated elasticities only account for sugary beverage expenditures within households and disregard expenses incurred outside the home. These expenditures may exhibit different patterns compared to those observed within households. Future research should consider the sensitivity of this type of demand and its potential impact on the results of SSB tax policies. The utilization of scanned data can help mitigate some of these limitations and contribute to generating further evidence. Lastly, limitations are also noticeable in the model's adjustment measures (a low R-squared in conjunction with good RMSE and MAE), warranting caution when interpreting the results.

## Supporting information

**S1 Table. Descriptive statistic of price and quantity acquired in households.**
(DOCX)

**S2 Table. Estimated probit model for the occurrence of consumption of each category of products in households.**
(DOCX)

**S3 Table. Goodness of fit measures for QUAIDS mode.**
(DOCX)

**S1 Dataset. Full dataset of the study.**
(CSV)

## Author Contributions

**Conceptualization:** Larissa Barbosa Cardoso.

**Methodology:** Auberth Henrik Venson, Larissa Barbosa Cardoso.

**Writing – original draft:** Auberth Henrik Venson, Larissa Barbosa Cardoso, Flaviane Souza Santiago, Kênia Barreiro de Souza, Renata Moraes Bielemann.

**Writing – review & editing:** Auberth Henrik Venson, Larissa Barbosa Cardoso, Flaviane Souza Santiago, Kênia Barreiro de Souza, Renata Moraes Bielemann.

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
