## [Decision Letter · Decision Letter 0]

27 Mar 2023

PONE-D-23-03690Demand Price Elasticity for Ready-To-Drink Sugar-Sweetened Beverages in BrazilPLOS ONE

Dear Dr. Santiago,

Thank you for submitting your manuscript to PLOS ONE. After careful consideration, we feel that it has merit but does not fully meet PLOS ONE’s publication criteria as it currently stands. Therefore, we invite you to submit a revised version of the manuscript that addresses the points raised during the review process.

We look forward to receiving your revised manuscript.

Kind regards,

Mohammed Al-Mahish

Academic Editor

PLOS ONE

Journal Requirements:

 "We are grateful for the financial support of he National Council for Scientific Development and Tecnológico (CNPq) for the development of research."

5. Please ensure that you include a title page within your main document. You should list all authors and all affiliations as per our author instructions and clearly indicate the corresponding author.

6. We note you have included a table to which you do not refer in the text of your manuscript. Please ensure that you refer to Table 3 in your text; if accepted, production will need this reference to link the reader to the Table.

Additional Editor Comments:

One reviewer recommended rejecting your paper while another reviewer recommended a minor revision. Thus, I have decided to give you a second chance to improve your paper by addressing reviewers comments as well as the following comments:

Make sure your paper meets PLOS ONE style including referencing styleThe estimated results of own price elasticity of water and milk showed that they are price elastic, which is not practically realistic. Thus, make sure to conduct diagnostic tests to asses the validity of your model. Also, report and discuss goodness of fit measures. You may consider omitting water and milk if the omission will not result in omitted variable bias.If you want to estimate price elasticity by income group as reported in table 8, you may consider using quantile regression.Please check the numbering of your tables. It seems tables 4-6 are missing.

Reviewers' comments:

Reviewer's Responses to Questions

**Comments to the Author**

1. Is the manuscript technically sound, and do the data support the conclusions?

Reviewer #1: No

Reviewer #2: Yes

2. Has the statistical analysis been performed appropriately and rigorously? 

Reviewer #1: No

Reviewer #2: Yes

3. Have the authors made all data underlying the findings in their manuscript fully available?

Reviewer #1: No

Reviewer #2: Yes

4. Is the manuscript presented in an intelligible fashion and written in standard English?

Reviewer #1: No

Reviewer #2: Yes

5. Review Comments to the Author

Reviewer #1: This study aims to estimate the price elasticity of demand for ready-to-drink sugar sweetened beverages in Brazil. The findings show that a taxation policy on ready-to-drink sweetened beverages has the potential to reduce the sugar consumption of the Brazilian population since an increase in the price of the product will lead to a more than proportional decrease in its demand.

Below are the major concerns for the authors to consider.

1. The aim and foremost contribution of this study is to estimate price elasticities. Unfortunately, there is no detail throughout the paper about their price data, not even any summary statistics. Household Budget/Expenditure Survey do not normally contain detailed individual household price data. It is not clear what prices they use in estimating QUAIDS. Chances are they probably used price indices which only vary along time but are invariant across households. If this is the case, compared to the detailed household consumption data, there is concerning lack of variation in the price data. Please see Hoderlein and Mihaleva (2008) for more details and solutions.

2. In Table 1, the sum of all Shares of Income Spent (%) in the last column is 99.98%, which means Brazilian households spent almost all their income on food. There was no spending on clothing, housing, health, education etc, which is not possible. I think what it means by Income in the paper is actually total Food expenditure. If this is the case, it is fine to call what they estimated expenditure elasticities, but cannot say much about how consumption of RTD SSBs is related to the household total budget/income by only looking at the estimated expenditure elasticities. It is possible that as household income increases, they might increas their share of spending more on luxuries, cars, holidays etc, leading to lower share of Food in the total budget. To really link consumption of SSBs to the income, assuming weakly separable household utility, how households allocate their total budget among more broad categories such as Food, Clothing, Education, Health etc, should be specified and estimated on top of the current demand system specified for the sub-categories within Food.

3. When using micro household level consumption data, how to properly deal with zero consumption to produce unbiased estimates is a very important issue, which has seen a large number of very significant studies in the literature by leading scholars. Unfortunately, in this paper, the authors failed to provide sufficient details as to how they deal with the zeros as a standalone paper. Only two minor references were provide, one of which is not in English making it very difficult for non-Spanish readers to follow. It is fine to model zero consumption decisions separately using probit; however, questions such as how the estimated probabilities were included in the estimation of the demand system, how the expected elasticities for the whole sample were derived and how they produced the corresponding standard errors, using Delta methods or Bootstrap, are unfortunately not clear at all.

Reference:

Hoderlein, S. and Mihaleva, S. (2008), ‘Increasing the Price Variation in a Repeated Cross Section’, Journal of Econometrics, 147, 316–25.

Reviewer #2: The paper discusses an interesting issue of policy relevance; Demand Price Elasticity for Ready-To-Drink Sugar-Sweetened Beverages in Brazil. While the paper is well delivered, I provide some comments for improvement.

1. The term ‘Demand Price Elasticity’ should be checked in title.

2. What are the contribution of the research to the literature, policy planners and consumers should add to the abstract.

3. Introduction needs some data of how much percentage of SSB consumption increased between 2009 and 2014 in Latin American countries. Similarly how much percentage of risk of developing obesity and diseases increased in?

4. How taxation policy impacted on price of sugar-sweetened beverages need some explanation for general reader in introduction.

5. Page 7 in Introduction paragraph 2 Drop in productivity of what?

6. Research objectives/research questions are not well defined, expenditure and income spent on SSB by type of SSB and subgroups are omitted to be belonging to the objectives.

7. The rationale and applicability of applying the Quadratic Almost Ideal Demand System (QUAIDS) methods needs to be further explained.

8. Where is the estimated probit models? Please give the equation with empirical model, and variable explanation is better placing in a table instead of description. And please also provide the steps where consumption probabilities were inserted into the estimation of the demand system and the subsequent calculation of elasticities.

9. Page 13, 2nd paragraph, what does it mean ‘24.05% of the families’? Is it surveyed families, please make it clear.

10. The data and analyses are well described.

11. In concluding section (last paragraph) please correlate the tax with price and add a strong policy implication based on findings.

12. Please provide the data of prices and annual demand (if possible, otherwise amount of consumption) for all SSB in 2017-2018 as supplementary.

13. Finally, the language needs further polishing.

6. PLOS authors have the option to publish the peer review history of their article (what does this mean?). If published, this will include your full peer review and any attached files.

Reviewer #1: No

Reviewer #2: **Yes: **Mst. Esmat Ara Begum (BARI034), Senior Scientific Officer, Bangladesh Agricultural Research Institute

---

## [Author Response · Author response to Decision Letter 0]

11 May 2023

Journal Requirements

Answer: We have done adjustments to attend the PLOS ONE style.

2. Please ensure that you include a title page within your main document. We do appreciate that you have a title page document uploaded as a separate file, however, as per our author guidelines (http://journals.plos.org/plosone/s/submission-guidelines#loc-title-page) we do require this to be part of the manuscript file itself and not uploaded separately. Could you therefore please include the title page into the beginning of your manuscript file itself, listing all authors and affiliations.

Answer: We have done adjustments to attend the PLOS ONE style. 

3. Thank you for stating the following financial disclosure: "We are grateful for the financial support of the National Council for Scientific Development and Tecnológico (CNPq) for the development of research." Please state what role the funders took in the study. If the funders had no role, please state: "The funders had no role in study design, data collection and analysis, decision to publish, or preparation of the manuscript."If this statement is not correct you must amend it as needed. Please include this amended Role of Funder statement in your cover letter; we will change the online submission form on your behalf.

Answer: We have done the adjustments to attend the PLOS ONE style. We amended Role of Funder statement in the cover letter.

Answer: Thank you for your comment. We will finalize the organization of the database and submit it soon.

5. Please ensure that you include a title page within your main document. You should list all authors and all affiliations as per our author instructions and clearly indicate the corresponding author.

Answer: We thank the comment. We have included the recommended information on the first page of the revised manuscript.

6. We note you have included a table to which you do not refer in the text of your manuscript. Please ensure that you refer to Table 3 in your text; if accepted, production will need this reference to link the reader to the Table.

Answer: We thank the comment. We adjusted the table numbering in the text.

Additional Editor comments

1. “Make sure your paper meets PLOS ONE style including referencing style”.

Answer: Suggestion accepted. We have done adjustments to attend the PLOS ONE style.

2. “The estimated results of own price elasticity of water and milk showed that they are price elastic, which is not practically realistic. Thus, make sure to conduct diagnostic tests to assess the validity of your model. Also, report and discuss goodness of fit measures. You may consider omitting water and milk if the omission will not result in omitted variable bias”.

Answer: We appreciate your comments and thank this opportunity to clarify the obtained result for water. The results indicate an elastic demand, similar to results obtained by Colchero et al. (2015), Guerrero-Lopes et al. (2017). Although our results show a higher magnitude, we understand that it would not be appropriate to exclude this category from the demand system. As the results obtained by Colchero et al. (2017) reveled, water serves as an immediate substitute for SSBs, and its demand increases with SSB taxation.

Colchero MA, Salgado JC, Unar-Munguía M, Hernández-Ávila M, Rivera-Dommarco JA. Price elasticity of the demand for sugar sweetened beverages and soft drinks in Mexico. Econ Hum Biol. 2015 Dec;19:129-37.

Guerrero-López, C.M., Unar-Munguía, M. & Colchero, M.A. Price elasticity of the demand for soft drinks, other sugar-sweetened beverages and energy dense food in Chile. BMC Public Health 17, 180 (2017).

Colchero MA, Molina M, Guerrero-López CM. After Mexico Implemented a Tax, Purchases of Sugar-Sweetened Beverages Decreased and Water Increased: Difference by Place of Residence, Household Composition, and Income Level. J Nutr. 2017 Aug;147(8):1552-155.

3. “If you want to estimate price elasticity by income group as reported in table 8, you may consider using quantile regression”.

Answer: Thank you for your valuable comment. Our intention was to estimate the elasticities by income subgroups. We agree that considering income quintiles is more appropriate in this case. We re-estimated the results, using the same methodology for the subsamples of the poorest households (first quintile) and the wealthiest households (fifth quintile). The results are presented in Table 5.

4. “Please check the numbering of your tables. It seems tables 4-6 are missing”.

Answer: Suggestion accepted. We adjusted the table numbering in the text.

Reviewer #1 

1. “The aim and foremost contribution of this study is to estimate price elasticities. Unfortunately, there is no detail throughout the paper about their price data, not even any summary statistics. Household Budget/Expenditure Survey do not normally contain detailed individual household price data. It is not clear what prices they use in estimating QUAIDS. Chances are they probably used price indices which only vary along time but are invariant across households. If this is the case, compared to the detailed household consumption data, there is concerning lack of variation in the price data. Please see Hoderlein and Mihaleva (2008) for more details and solutions”.

Answer: We appreciate your comments, and we are grateful for the opportunity to provide clarification about the prices used. The information contained in the database refers to the purchases made by each household over seven consecutive days. The database does not include information on the prices paid by consumers. We calculated the unit value by dividing the total expenditure by the quantity acquired, which was considered as a proxy for the paid price. For the estimation of the demand system, the average price of each category was considered. information was used to calculate the Matsuda price index and Tornqvist price index, which were utilized in the demand system. We have incorporated these information in the text, specifically within the section that describes the variables and in the method description.

2. “In Table 1, the sum of all Shares of Income Spent (%) in the last column is 99.98%, which means Brazilian households spent almost all their income on food. There was no spending on clothing, housing, health, education etc, which is not possible. I think what it means by Income in the paper is actually total Food expenditure. If this is the case, it is fine to call what they estimated expenditure elasticities, but cannot say much about how consumption of RTD SSBs is related to the household total budget/income by only looking at the estimated expenditure elasticities. It is possible that as household income increases, they might increas their share of spending more on luxuries, cars, holidays etc, leading to lower share of Food in the total budget. To really link consumption of SSBs to the income, assuming weakly separable household utility, how households allocate their total budget among more broad categories such as Food, Clothing, Education, Health etc, should be specified and estimated on top of the current demand system specified for the sub-categories within Food”.

Answer: Thank you for your comment. The database has some limitations, as for certain consumption categories, there is only information on expenditure. In these cases, there is no record of the quantity acquired, which limits the inclusion of other expense groups in the demand system. Taking this into account, we chose to calculate the demand elasticity considering food expenditures. We have taken your suggestion and changed the terms related to income elasticity to expenditure elasticity. Additionally, we have removed any mentions in the text regarding the relationship between income and RTB SBB.

3. “When using micro household level consumption data, how to properly deal with zero consumption to produce unbiased estimates is a very important issue, which has seen a large number of very significant studies in the literature by leading scholars. Unfortunately, in this paper, the authors failed to provide sufficient details as to how they deal with the zeros as a standalone paper. Only two minor references were provided, one of which is not in English making it very difficult for non-Spanish readers to follow. It is fine to model zero consumption decisions separately using probit; however, questions such as how the estimated probabilities were included in the estimation of the demand system, how the expected elasticities for the whole sample were derived and how they produced the corresponding standard errors, using Delta methods or Bootstrap, are unfortunately not clear at all”.

Reference:

Hoderlein, S. and Mihaleva, S. (2008), ‘Increasing the Price Variation in a Repeated Cross Section’, Journal of Econometrics, 147, 316–25.

Answer: We appreciate the comments. For the elasticity estimates presented, we consider the issue of zero consumption and endogeneity problem of total expenditure on food. To address the problem of zero consumption, we adopted a two-stage procedure. In the first stage, the probability of consumption for each category was estimated using a probit model. The results are presented in Table S1 in the appendix. In the second stage, we calculated the cumulative distribution functions (CDF) and probability density functions (PDF), which were included in the demand system. Additionally, we conducted a regression analysis using a set of independent variables that represent household characteristics and calculated the residuals of the regression. The residuals were included as independent variable in the QUAIDS model. We have provided detailed information on these aspects in the statistical method section and included the equations for calculating the elasticities.

Reviewer 2

1. “The term ‘Demand Price Elasticity’ should be checked in title”.

Answer: Thank you for your comment. The title was change.

“ Price Elasticity of Demand for Ready-To-Drink Sugar-Sweetened Beverages in Brazil”

2. “What are the contribution of the research to the literature, policy planners and consumers should add to the abstract”.

Answer: Thank you for your comment. We include this information in the abstract.

“The present study advances the literature by proposing a breakdown between ready-to-drink sugar-sweetened beverages and sugar-sweetened beverages that require some preparation before being consumed. With this disaggregation, it is possible to obtain more accurate elasticities for the group of products that will be effectively taxed. Thus, public policies can be directed to reduce the sugar consumption of the Brazilian population and minimize health risks”.

3. “Introduction needs some data of how much percentage of SSB consumption increased between 2009 and 2014 in Latin American countries. Similarly, how much percentage of risk of developing obesity and diseases increased in?”

Answer: Suggestion accepted. The Introduction has been revised to provide a clearer explanation of the study's motivation and rationale. We have included data on the consumption of sweetened beverages in both Latin America and Brazil. Furthermore, we have expanded the discussion to include information on non-communicable diseases (NCDs), recognizing the link between sweetened beverages and various health conditions beyond just obesity.

4. “How taxation policy impacted on price of sugar-sweetened beverages need some explanation for general reader in introduction”.

Answer: Suggestion accepted. We have included the rationale for taxing SSBs in the introduction. 

5. “Page 7 in Introduction paragraph 2 Drop in productivity of what?”

Answer: Thank you for bringing that to our attention. In this part we referred to loss of productivity due to premature deaths. We have addressed this point by providing clarification in the text.

6. “Research objectives/research questions are not well defined, expenditure and income spent on SSB by type of SSB and subgroups are omitted to be belonging to the objectives”.

Answer: Thank you for your comment. We have adjusted this in the text.

7. “The rationale and applicability of applying the Quadratic Almost Ideal Demand System (QUAIDS) methods needs to be further explained”.

Answer: Thank you your comment. We attended this suggestion, including additional information of QAIDS in methods section.

8. “Where is the estimated probit models? Please give the equation with empirical model, and variable explanation is better placing in a table instead of description. And please also provide the steps where consumption probabilities were inserted into the estimation of the demand system and the subsequent calculation of elasticities”.

Answer: We appreciate your comment. We attended this suggestion, including additional information of in methods section. Additionally, we present the probit model results in Table S1 in the appendix.

9. “Page 13, 2nd paragraph, what does it mean ‘24.05% of the families’? Is it surveyed families, please make it clear”.

Answer: Thank you for your observation. We adjust the text to make clear that we refer to the percentage of surveyed households.

10. “The data and analyses are well described”.

Answer: Thank you for your comment.

11. “In concluding section (last paragraph), please correlate the tax with price and add a strong policy implication based on findings”.

Answer: Suggestion accepted. We included one last paragraph in the concluding section.

“Considering that, a taxation on sweetened beverages can contribute to improving health outcomes by increasing the price of these beverages and reducing consumption. With a higher sensitivity among lower-income households, the taxation of SSBs has the potential to promote health equity, especially in terms of the prevalence of non-communicable chronic diseases.”

12. “Please provide the data of prices and annual demand (if possible, otherwise amount of consumption) for all SSB in 2017-2018 as supplementary”.

Answer: Thank you for your comment. We have included in the appendix a table with descriptive statistics of the prices. Please let us know if this information is sufficient to attend your suggestion.

13. “Finally, the language needs further polishing”.

Answer: Thank you for your comment. We accepted and attended your suggestion rewritten the text.

---

## [Decision Letter · Decision Letter 1]

13 Jun 2023

PONE-D-23-03690R1Price elasticity of demand for ready-to-drink sugar-sweetened beverages in BrazilPLOS ONE

Dear Dr. Santiago,

Thank you for submitting your manuscript to PLOS ONE. After careful consideration, we feel that it has merit but does not fully meet PLOS ONE’s publication criteria as it currently stands. Therefore, we invite you to submit a revised version of the manuscript that addresses the points raised during the review process.

ACADEMIC EDITOR:

Although the paper has been improved, one of the reviewers has asked for a major correction on your model. Also, the necessary model’s diagnostic tests and goodness of fit measures are still missing.Thus, I invite you for a second revision to address those concerns.

We look forward to receiving your revised manuscript.

Kind regards,

Mohammed Al-Mahish

Academic Editor

PLOS ONE

Reviewers' comments:

Reviewer's Responses to Questions

**Comments to the Author**

1. If the authors have adequately addressed your comments raised in a previous round of review and you feel that this manuscript is now acceptable for publication, you may indicate that here to bypass the “Comments to the Author” section, enter your conflict of interest statement in the “Confidential to Editor” section, and submit your "Accept" recommendation.

Reviewer #1: (No Response)

2. Is the manuscript technically sound, and do the data support the conclusions?

Reviewer #1: Yes

3. Has the statistical analysis been performed appropriately and rigorously? 

Reviewer #1: Yes

4. Have the authors made all data underlying the findings in their manuscript fully available?

Reviewer #1: No

5. Is the manuscript presented in an intelligible fashion and written in standard English?

Reviewer #1: Yes

6. Review Comments to the Author

Reviewer #1: The authors have done a great job addressing most of my previous comments. For this round, I only have concerns about the prices they use for estimation. In the revised draft, they use unit values as individual prices, which is worrisome. In estimating demand system, you want to avoid using unit values as possible as you can since they have quality effects embedded in them and thus are correlated with preferences making them endogenous. For example, for individuals who tend to buy premium RTD SSBs versus others who buy average brands, the former would pay much higher unit values than the latter for the same amount of SSBs consumed, reflecting individual's preferences. Please see Nelson (1991) and Nelson (1990) that discuss the issue rather well.

One approach you might want to consider in order to fix this issue is to look at those sub-products within the aggregate group of RTD SSBs, such as sodas, soft drinks, nectars etc. Calculate unit values for these sub-products and use these sub-group unit values to construct a Laspeyres or Paasche type index for the aggregate group RTD SSBs. The idea is to start with the most elementary prices in the data set and construct the indexes for the aggregates. It will be less of a problem if you start with the disaggregate unit values and construct index numbers based on them the aggregate group of RTD SSBs.

Reference:

Nelson, J. A. (1991). Quality variation and quantity aggregation in consumer demand for food. American Journal of Agricultural Economics, 73(4), 1204-1212.

Nelson, J. A. (1990). Quantity aggregation in consumer demand analysis when physical quantities are observed. The Review of Economics and Statistics, 153-156.

7. PLOS authors have the option to publish the peer review history of their article (what does this mean?). If published, this will include your full peer review and any attached files.

Reviewer #1: No

---

## [Author Response · Author response to Decision Letter 1]

17 Aug 2023

Dear editor,

We would like to thank you for the time spending evaluating our paper. The modifications are highlighted in the article (file labeled 'Revised Manuscript with Track Changes'). 

Editor comment

Although the paper has been improved, one of the reviewers has asked for a major correction on your model. Also, the necessary model’s diagnostic tests and goodness of fit measures are still missing. Thus, I invite you for a second revision to address those concerns.

Answer: Thanks for your comment. We have done the corrections in our model, as proposed by the reviewer, and present the results with the adjustments. Also, we present table S3 with goodness of fit measures, the table shows that the model presented a good quality of fit, with low values for RSME and MAE, and good values of R2, considering the characteristics of the database, with F test indicating global significance of all equations of the system, information criteria were also reported.

Reviewer 1 Comment

Reviewer #1: The authors have done a great job addressing most of my previous comments. For this round, I only have concerns about the prices they use for estimation. In the revised draft, they use unit values as individual prices, which is worrisome. In estimating demand system, you want to avoid using unit values as possible as you can since they have quality effects embedded in them and thus are correlated with preferences making them endogenous. For example, for individuals who tend to buy premium RTD SSBs versus others who buy average brands, the former would pay much higher unit values than the latter for the same amount of SSBs consumed, reflecting individual's preferences. Please see Nelson (1991) and Nelson (1990) that discuss the issue rather well.

One approach you might want to consider in order to fix this issue is to look at those sub-products within the aggregate group of RTD SSBs, such as sodas, soft drinks, nectars etc. Calculate unit values for these sub-products and use these sub-group unit values to construct a Laspeyres or Paasche type index for the aggregate group RTD SSBs. The idea is to start with the most elementary prices in the data set and construct the indexes for the aggregates. It will be less of a problem if you start with the disaggregate unit values and construct index numbers based on them the aggregate group of RTD SSBs.

Answer: Thanks for your comment. We really appreciate it. We have done the quality-adjustment in the unit values. To do this, we follow the suggested strategy to use indexes for the aggregated category and calculated them as the expenditure divided by total quantity consumed in the category, weighted by expenditure share. Also, we clarify how to we have treated the missing values for non-consuming households. Please, see the changes in lines 219-224. The new results with the quality-adjusted unit value are shown in Results section. 

Do not hesitate to contact us if any other information is needed at this stage of the process.

Best regards,

Authors

---

## [Editor Report · Decision Letter 2]

30 Aug 2023

PONE-D-23-03690R2Price elasticity of demand for ready-to-drink sugar-sweetened beverages in BrazilPLOS ONE

Dear Dr. Santiago,

Thank you for submitting your manuscript to PLOS ONE. After careful consideration, we feel that it has merit but does not fully meet PLOS ONE’s publication criteria as it currently stands. Therefore, we invite you to submit a revised version of the manuscript that addresses the points raised during the review process.

ACADEMIC EDITOR:Please discuss the limitation of your study in the conclusion by focusing on the weaknesses of your study such as positive own price elasticity of some items, low R-squared value…etc. Also, provide suggestions and recommendation for future research.

We look forward to receiving your revised manuscript.

Kind regards,

Mohammed Al-Mahish, Ph.D.

Academic Editor

PLOS ONE
---

## [Author Response · Author response to Decision Letter 2]

10 Oct 2023

Dear editor,

We would like to thank you for the time spending evaluating our paper. The modifications are highlighted in the article (file labeled 'Revised Manuscript with Track Changes'). 

Editor comment

Please discuss the limitation of your study in the conclusion by focusing on the weaknesses of your study such as positive own price elasticity of some items, low R-squared value…etc. Also, provide suggestions and recommendation for future research.

Answer: Thanks for your comment. Your recommendations and insights were extremely helpful in enhancing our work. All suggested changes have been incorporated into the revised article. We believe these alterations have significantly strengthened the work, making it more robust and relevant to Plos One's readership. Please, see the changes in lines 471-486.

Please do not hesitate to reach out to us should there be any additional information or clarification needed.

Sincerely,

The authors

---

## [Editor Report · Decision Letter 3]

12 Oct 2023

Price elasticity of demand for ready-to-drink sugar-sweetened beverages in Brazil

PONE-D-23-03690R3

Dear Dr. Santiago,

We’re pleased to inform you that your manuscript has been judged scientifically suitable for publication and will be formally accepted for publication once it meets all outstanding technical requirements.

Kind regards,

Mohammed Al-Mahish, Ph.D.

Academic Editor

PLOS ONE

Additional Editor Comments (optional):

Comprehensive English proofreading inspection by a professional English proofreader is highly recommended for your paper.
---

## [Editor Report · Acceptance letter]

23 Oct 2023

PONE-D-23-03690R3 

Price elasticity of demand for ready-to-drink sugar-sweetened beverages in Brazil 

Dear Dr. Santiago:

I'm pleased to inform you that your manuscript has been deemed suitable for publication in PLOS ONE. Congratulations! Your manuscript is now with our production department. 

Kind regards, 

on behalf of

Dr. Mohammed Al-Mahish 

Academic Editor

PLOS ONE